# Comparative Evaluation of [^68^Ga]Ga-Fibroblast Activation Protein Inhibitor vs. [^18^F]FDG as a Novel Radiotracer for Biology-Guided Image Radiotherapy

**DOI:** 10.3390/cancers17223648

**Published:** 2025-11-13

**Authors:** Lin Qiu, Yue Chen, Trevor Ketcherside, Zhixing Wang, Todd DeWees, Terence M. Williams, Arya Amini, Sagus Sampath, Scott Glaser, Yi-Jen Chen, Liu Lin, David Leung, An Liu, Heather M. McGee

**Affiliations:** 1Department of Nuclear Medicine, Affiliated Hospital of Southwest Medical University and Institute of Nuclear Medicine, Southwest Medical University, Luzhou 646000, China; qiulin17111210041@163.com (L.Q.); chenyue5523@126.com (Y.C.); linliu816@163.com (L.L.); 2Department of Radiation Oncology, City of Hope National Medical Center, Duarte, CA 91010, USA; tketcherside@coh.org (T.K.); todd.dewees@gmail.com (T.D.); terwilliams@coh.org (T.M.W.); aamini@coh.org (A.A.); ssampath@coh.org (S.S.); sglaser@coh.org (S.G.); yichen@coh.org (Y.-J.C.); 3RefleXion Medical, Inc., Hayward, CA 94545, USA; davidleungmd@gmail.com

**Keywords:** biology guided radiotherapy (BgRT), image-guided adaptive radiation therapy, fibroblast activation protein inhibitor, gallium-68

## Abstract

Biology-guided radiotherapy (BgRT) uses PET radiotracer emissions as image guidance to deliver adaptive targeted radiation to tumors. Due to limitations of the PET radiotracer ^18^F-Fluorodeoxyglucose ([^18^F]FDG), there is significant interest in developing alternative radiotracers for BgRT. Fibroblast activation protein (FAP) is a transmembrane protein that is overexpressed in cancer-associated fibroblasts within multiple tumor types with minimal expression in normal tissues. Fibroblast activation protein inhibitors (FAPIs) bind avidly to FAP and can be labeled with a radiotracer such as Gallium-68 (^68^Ga). Multiple studies have compared [^68^Ga]Ga-FAPI-04 vs. [^18^F]FDG for diagnostic imaging, but no one has compared the utility of [^68^Ga]Ga-FAPI-04 vs. [^18^F]FDG for BgRT. This study was designed to compare calculated parameters from [^68^Ga]Ga-FAPI-04 vs. [^18^F]FDG PET-CT to determine if FAPI-04 can be a useful biological fiducial for BgRT. The findings from this work could have a significant impact on the field by providing evidence that [^68^Ga]Ga-FAPI-04 is a superior agent for image-guided adaptive radiation therapy.

## 1. Introduction

Biology-guided radiotherapy (BgRT) continuously acquires PET signals to provide real-time image-guided radiation delivery [1]. This method may facilitate the treatment of oligometastatic or oligoprogressive lesions based on their PET radiotracer avidity, thus providing a novel motion management approach in which the PET signal acts as a fiducial for radiation treatment [2,3,4]. [^18^F]FDG is the most commonly used radiotracer for PET-CT. However, there are multiple limitations to using [^18^F]FDG as a radiotracer for BgRT. [^18^F]FDG is not specific for tumor cells and is taken up by all proliferating cells, resulting in significant background in the brain, liver, and bladder, and is a glucose analog, so patients must fast before imaging to ensure maximal tumor uptake [5].

There is significant interest in developing new radiotracers in both the diagnostic setting (for PET imaging) and the therapeutic setting (as a systemically injected radiopharmaceutical). Fibroblast activation protein (FAP) is a type II transmembrane serine protease that is overexpressed in cancer-associated fibroblasts (CAFs) found in most epithelial tumors [6]. Multiple groups have developed radionuclide-labeled fibroblast activation protein inhibitors (FAPIs) that specifically bind to FAP for diagnostic and therapeutic purposes [7,8].

FAPI-PET-CT can image many different tumors (including pancreatic, lung, esophageal, breast, and gastric cancers) [9,10]. These radionuclide-labeled FAPI compounds have less uptake in the brain and liver, allowing for more accurate tumor delineation with less background. In addition, FAPI PET-CT is better than [^18^F]FDG PET-CT for detecting metastatic lymph nodes and distant metastases [6,11]. While there are multiple studies comparing characteristics of FAPI vs. [^18^F]FDG for diagnostic imaging [12,13], there has never been a comparison of FAPI vs. [^18^F]FDG in fulfilling the prerequisites for the delivery of BgRT.

Gallium-68 [^68^Ga]FAPI-04 is a well-studied quinoline-based radiolabeled compound targeting FAP with very low physiologic background uptake and good stability in serum that is quickly removed from normal organs in vivo. The chemical structures of [^68^Ga]Ga-FAPI-04 and [18F]FDG are shown in the Appendix A. Since [^68^Ga]Ga-FAPI-04 PET can be performed without fasting and has a short uptake time, it has the potential to improve patient comfort and accelerate clinical workflow. In this study, we sought to quantitatively assess the utility of [^68^Ga]Ga-FAPI-04 vs. [^18^F]FDG for biology-guided radiotherapy by comparing two key characteristics of radiotracers, the Normalized Net Activity Concentration (NNA) (goal > 5 kBq/mL to deliver BgRT) and the Normalized Target Signal (NTS) (goal > 2.7 for BgRT treatment planning & > 2.0 for BgRT delivery) [14]. To do this, we compared and analyzed PET imaging datasets of patients with a diverse group of solid tumors imaged with both [^18^F]FDG and [^68^Ga]Ga-FAPI-04.

## 2. Materials and Methods

This study was conducted under IRB 20039. PET-CTs were obtained for 50 patients with liver, pancreas, lung, head and neck, and cervical cancers using both [^18^F]FDG and [^68^Ga]Ga-FAPI-04 radiotracers (*n* = 10 patients per disease site with 4 DICOM image sets per patient (PET and CT for both [^18^F]FDG and [^68^Ga]Ga-FAPI-04). A radiation oncologist with expertise in each disease site contoured the gross tumor volume (GTV) on both the [^18^F]FDG PET-CT images and [^68^Ga]Ga-FAPI-04 PET-CT images. In addition, auto-contours were generated on the FDG PET-CT and FAPI PET-CT images using an auto-threshold of 40% of the maximum Standardized Uptake Value (SUV) for each lesion using Varian Velocity^TM^ version 3.2 (Figure 1A).

We chose to use a 40% SUVmax threshold for both [^68^Ga]Ga-FAPI-04 and [^18^F]FDG auto-contouring because this is the most common maximum standardized uptake value (SUVmax) used to delineate the GTV using FDG [15]. While [^68^Ga]Ga-FAPI-04 PET has not yet been used in radiation treatment planning, a 40% SUVmax threshold has been used to compare tumor volume between [^68^Ga]Ga-FAPI-04 and [^18^F]FDG [16,17].

To define the BgRT planning tumor volume (PTV), a margin was added to the GTV (3 mm for head and neck cancer and 5 mm expansion for other disease sites). A customized “GTV motion envelope” was added to the BgRT PTV, based on the typical extent of motion observed on 4D-CT scans for each tumor site. The Biology Tracking Zone (BTZ) was defined as the Internal Target Volume (ITV) plus the biological-guidance margin with an added setup margin (Figure 1B). ITV accounts for tumor motion during respiration and is derived from 4D-CT or PET imaging to ensure adequate coverage during radiation treatment. For gastrointestinal tumors near the diaphragm (i.e., pancreatic and liver tumors), it was assumed that these tumors move about 1 cm in the superior-inferior direction and 5 mm in the anterior–posterior direction. For lung cancer, it was assumed that they move uniformly about 1 cm in all directions. For head and neck and cervical tumors, it was assumed that these tumors have minimal (0.2–0.3mm) movement in all directions. The following parameters were calculated for both physician contours and auto-contours: GTV (cc), SUVmax, volume of GTV correlating with 80% of SUVmax, and SUVmean.

To assess the feasibility of biology-guided radiotherapy (BgRT), two unique imaging-derived metrics: Normalized Net Activity concentration (NNA) and Normalized Target Signal (NTS) are required by the FDA to determine whether a radiotracer provides sufficient signal intensity within the tumor (NNA) and adequate contrast relative to surrounding tissue (NTS) to enable BgRT planning and delivery. NNA reflects the absolute radiotracer emission within the target volume, while NTS quantifies the tumor-to-background signal ratio [18,19,20]. Both metrics are essential for ensuring that BgRT systems can reliably track and treat tumors in real time.

The activity concentration of the target was defined as Mean AC (kBq/mL) in the volume of GTV receiving 80% of SUV max. The activity concentration of the Background was defined as mean AC (kBq/mL) in the 3 mm ring around the BTZ. The standard deviation (STD) of the Activity Concentration_Background was measured using Eclipse. The normalized net activity (NNA) Concentration was defined as AC_Target minus AC_Background and corrected by time decay due to individual patient injection time variation. Time decay correction was calculated based on (t_elapsed_−t_BgRT_) and the half-life of the radiotracer (68.3 min for [^68^Ga]Ga-FAPI-04 and 109 min for [^18^F]FDG), where t_elapsed_ is the time from injection to image acquisition and t_BgRT_ is estimated as 60 min for [^18^F]FDG BgRT treatment and 30 min for [^68^Ga]Ga-FAPI-04. NNA FAPI was further normalized by the ratio of injected activity of [^18^F]FDG to [^68^Ga]Ga-FAPI-04 because the amount of [^18^F]FDG injected compared to [^68^Ga]Ga-FAPI-04 was significantly higher, as shown in Equation (1) and Equation (2), respectively. The NTS was calculated as follows: (AC_Target−AC_Background)/Standard Deviation of Background (Figure 1C). At the outset of the study, we determined that the Normalized Net Activity Concentration in the target must be >5 kBq/mL to treat patients with [^18^F]FDG-directed BgRT. In addition, we determined that the Normal Target Signal needs to be >2.7 for BgRT treatment planning and >2.0 for BgRT delivery. Descriptive statistics with 95% confidence intervals were calculated for all parameters. Two-sided paired *t*-tests were utilized to test for significant differences (α = 0.05) between [^68^Ga]Ga-FAPI-04 and [^18^F]FDG for SUV mean, NNA, and NTS.


(1)
$$NNA_{FAPI}\;=\;({AC}_{Target}-{AC}_{Background})*\frac{FDG_{injected\;activity}}{FAPI_{injected\;activity}}\;*\;e^{0.693\;*\;\frac{t_{elapsed}-t_{BgRT}}{68.3}}$$



(2)
$$NNA_{FDG}\;=\;(AC_{Target}\;-\;AC_{Background})\;*\;e^{0.693*\;\frac{t_{elapsed}\;-\;t_{BgRT}}{109}}$$


## 3. Results

### 3.1. Normalized Net Activity Concentration (NNA) and Normalized Target Signal (NTS)

The normalized net activity concentration (NNA) was greater than 5.0 kBq/mL for all tumor sites using both [^18^F]FDG and [^68^Ga]Ga-FAPI-04, and the normalized target signal (NTS) was greater than 2.7 for all tumor types using both [^18^F]FDG and [^68^Ga]Ga-FAPI-04. This suggests that both radiotracers generate a sufficient signal-to-background ratio necessary to deliver BgRT (Figure 2A and Figure 3A). Paired *t*-tests were utilized to test for significant differences (α = 0.05) between [^68^Ga]Ga-FAPI-04 and [^18^F]FDG (Figure 2B,C, and Figure 3B,C). In order to compare the magnitude of difference between NTS_FAPI_ and NTS_FDG_, the t-ratio (which is the standardized mean difference in NTS_FAPI_ vs. NTS_FDG_) was also calculated for all disease sites (Figure 4B).

### 3.2. Gross Tumor Volume (GTV) Analysis

We conducted a volumetric analysis of the difference in GTV (cc) from physician contours on both FAPI and FDG PET-CT (Figure 5C), utilizing the paired *t*-test. The difference in GTV was statistically significant in pancreatic cancer, suggesting that [^68^Ga]Ga-FAPI-04 may identify stromal elements of pancreatic tumors that do not light up on [^18^F]FDG PET-CT. There were only small differences in the GTV for all other disease sites (lung, liver, H&N, and cervical cancers) (Figure 4C). Since FAPI staining was visualized in a rim around the central necrotic core for some liver lesions, some contoured GTVs had large volumes (>100 cc), which led to significant differences in the GTVs for [^68^Ga]Ga-FAPI-04 vs. [^18^F]FDG. It is challenging to deliver BgRT to tumors that do not have a homogeneous FAPI signal because the radiotracer signal is necessary for targeting the radiation. Therefore, liver lesions were excluded from the volumetric comparison because further investigation is needed to determine if [^68^Ga]Ga-FAPI-04 is useful for tumors in the liver.

#### 3.2.1. Pancreatic Cancer

Our comparison of NNA and NTS for [^68^Ga]Ga-FAPI-04 and [^18^F]FDG revealed differences in the applicability of BgRT for different tumor types. For pancreatic cancer, the mean NNA_FAPI_ was more than double the mean NNA_FDG_ (55.98 and 20.8, respectively; *p* = 0.0007). The mean NTS_FAPI_ is more than double the mean NTS_FDG_ (20.3 and 8.43, respectively; *p* = 0.0043). On average, there is approximately one more pancreatic tumor identified on [^68^Ga]Ga-FAPI-04 images than on [^18^F]FDG images (*p* = 0.0078). Based on the t-ratio describing the change in NTS between [^68^Ga]Ga-FAPI-04 and [^18^F]FDG, pancreatic cancer may be the most promising disease site to treat with FAPI BgRT.

#### 3.2.2. Lung Cancer

For lung cancer, the GTV, SUVmax, and SUVmean for [^68^Ga]Ga-FAPI-04 were all greater than for [^18^F]FDG. The mean NNA_FAPI_ was 46.39 kBq/mL, and the mean NNA_FDG_ was 34.57 kBq/mL (not significant). However, the mean NTS_FAPI_ was 34.11, and the mean NTS_FDG_ was 17.75 (*p* = 0.0036). Based on the t-ratio describing the change in NTS between [^68^Ga]Ga-FAPI-04 and [^18^F]FDG, lung cancer deserves further investigation as a disease site to treat with [^68^Ga]Ga-FAPI-04-based BgRT. For head and neck cancer, the mean NNA_FAPI_ was 56.77 kBq/mL and the mean NNA_FDG_ was 28.42 kBq/mL (*p* = 0.0058). The mean NTS_FAPI_ was 18.51, and the mean NTS_FDG_ was 15.48; the difference was not statistically significant. In addition, more lesions were found on [^68^Ga]Ga-FAPI-04-PET than on [^18^F]FDG -PET. Specifically, more lesions were identified in the lymph nodes, as has been described in the literature [12].

#### 3.2.3. Liver Cancer

For liver cancer, the mean NNA_FAPI_ was 43.41 kBq/mL and the mean NNA_FDG_ was 23.15 kBq/mL, *p* = 0.0605). The mean NTS_FAPI_ was 20.85, and the mean NTS_FDG_ was 7.18, which was statistically significant (*p* = 0.0112), suggesting that the NTS_FAPI_ for liver cancer was more than double the NTS_FDG_. Based on the t-ratio describing the change in NTS between [^68^Ga]Ga-FAPI-04 and [^18^F]FDG, liver cancer deserves further investigation for BgRT. Importantly, [^68^Ga]Ga-FAPI-04 has no background uptake in the liver like [^18^F]FDG. However, 4 of 10 liver lesions had a ring of FAPI signal around a central necrotic core (Figure 5A,B). Given that a relatively homogeneous FAPI signal is needed to target the whole tumor with BgRT, inhomogeneities of the [^68^Ga]Ga-FAPI-04 signal in liver lesions may exclude some patients with liver cancer from [^68^Ga]Ga-FAPI-04-based BgRT. In addition, the unusual pattern of [^68^Ga]Ga-FAPI-04 staining in a rim-like pattern led to some liver tumors having very large FAPI volumes.

#### 3.2.4. Cervical Cancer

For cervical cancer, the mean NNA_FAPI_ was 58.36 kBq/mL and the mean NNA_FDG_ was 37.81 kBq/mL, and the difference was statistically significant (*p* = 0.042). The mean NTS_FAPI_ was 4.18, and the mean NTS_FDG_ was 5.02, but the difference was not statistically significant. In addition, [^68^Ga]Ga-FAPI-04 has a strong signal in the endometrium due to stromal cells in the endometrium (Figure 5C,D) [11]. Therefore, the SUVmean of cervical cancer was similar to the SUVmean of the uterus (Figure 4), and this excessive background signal will likely preclude treatment of this cervical cancer with [^68^Ga]Ga-FAPI-04 BgRT.

## 4. Discussion

Here, we present the first study evaluating a quinoline-based FAP inhibitor bound to a DOTA chelator [^68^Ga]Ga-FAPI-04 versus [^18^F]FDG for real-time PET-based biology-guided radiation therapy in a variety of cancer types, including pancreatic cancer, liver cancer, lung cancer, head and neck, and cervical cancer. The goal of the study was to determine which disease sites could be treated with [^68^Ga]Ga-FAPI-04 BgRT more effectively than with [^18^F]FDG BgRT using established parameters required for the delivery of BgRT.

At the outset of this study, we chose to use the established thresholds of [^18^F]FDG-directed BgRT, namely that the NNA must be greater than 5.0 kBq/mL and the NTS must be greater than 2.7 kBq/mL for the delivery of BgRT. It should be noted, however, that the actual thresholds for these metrics have not yet been established for [^68^Ga]Ga-FAPI-04-directed BgRT. Furthermore, these threshold values are expected to be higher than those required for [^18^F]FDG due to the shorter half-life, smaller positron fraction, and higher maximum positron energy of 68Ga compared to 18F. Nevertheless, the NNA_FAPI_ was statistically significantly greater than the NNA_FDG_ in pancreatic cancer, head and neck cancer, and cervical cancer, suggesting that [^68^Ga]Ga-FAPI-04 reactivity is higher than [^18^F]FDG reactivity in these cancer types (Figure 2). This highlights the abundance of cancer-associated fibroblasts in these disease sites and emphasizes the fact that the tumor stroma makes up a large portion of these tumors. Not surprisingly, some of these tumor types with abundant stroma have previously been shown to be optimal targets for FAPI imaging [21,22]. NNA is a threshold value (or a binary variable), and values greater than 5 kBq/mL are needed to treat tumors with BgRT. In the majority of cases, NNA was greater than 5 kBq/mL, and therefore those patients may be suitable for treatment with BgRT using both [^68^Ga]Ga-FAPI-04 and [^18^F]FDG (as shown in Figure 4A). However, in disease sites where NNA_FAPI_ is significantly greater than NNA_FDG_, the BgRT treatment delivery may be more robust because of the stronger [^68^Ga]Ga-FAPI-04 PET signal, or we may be able to reduce the injected activity of [^68^Ga]Ga-FAPI-04 and use fewer millicuries for each treatment.

Next, we evaluated the Normalized Target Signal (NTS), which measures the difference in activity concentration between the target and background, divided by the standard deviation of the background, to capture the “signal to noise” ratio of each radiotracer. NTS is a continuous variable where larger values represent a greater signal-to-noise ratio, which is essential for accurate delivery of BgRT. As such, we consider NTS to be the most important variable to consider when evaluating radiotracers for BgRT. In this study, we found that the NTS_FAPI_ was greater than the NTS_FDG_ in pancreatic cancer, liver cancer, and lung cancer (Figure 3). In order to compare the magnitude of difference between NTS_FAPI_ and NTS_FDG_, we calculated the t-ratio (the standardized mean difference in NTS_FAPI_ vs. NTS_FDG_) (Figure 4B). Based on the t-ratio for change in NTS between [^68^Ga]Ga-FAPI-04 and [^18^F]FDG, we identified that pancreatic cancer, liver cancer, and lung cancer are all disease sites that deserve further investigation for [^68^Ga]Ga-FAPI-04-based BgRT.

The unusual distribution of [^68^Ga]Ga-FAPI-04 in some liver lesions (a rim of [^68^Ga]Ga-FAPI-04 around a central necrotic core) is as elucidated an interesting biological phenomenon that deserves further exploration. Further studies are needed to compare the [^68^Ga]Ga-FAPI-04 distribution in hepatocellular carcinoma (HCC) with that of colorectal metastases to the liver and visualize cancer-associated fibroblast-mediated crosstalk with HCC or metastatic cancer cells in the liver tumor microenvironment [23]. The lack of a significant difference in NNA between tracers in the liver and lung is worth further investigation. One possible explanation may be attributed to physiological background uptake: FDG has known hepatic and pulmonary uptake, while FAPI may show variable uptake depending on fibrosis and stromal-immune interactions. Another possible reason is the tumor microenvironment: Lesions in the liver and lung may have similar FAP expression or lytic activity as the organs themselves, reducing tracer contrast. Taken together, our data suggest that a subset of patients with liver tumors may benefit from BgRT, but careful patient selection will be required due to the spatial distribution of [^68^Ga]Ga-FAPI-04 in liver tumors.

In addition, we noted that [^68^Ga]Ga-FAPI-04 has better uptake in the lymph nodes than [^18^F]FDG in lung cancer, head and neck cancer, and cervical cancer. This is consistent with published reports showing that another ^68^Ga-radiolabelled FAPI was useful to detect nodal metastases in four out of five nodes that were confirmed pathologically [24]. Since CAFs promote tumor growth and invasion, the presence of CAFs correlates with a poor prognosis [21,23,24], and there may be more [^68^Ga]Ga-FAPI-04-avid regions in advanced-stage tumors that have more nodal involvement.

Lastly, we observed that there is a positive [^68^Ga]Ga-FAPI-04 signal in the normal uterus, likely related to high stromal content within the endometrium or myometrium [11]. This created a significant background signal that may interfere with BgRT treatment for cervical cancer.

Another caveat to consider is that the expression of FAPI is not limited to cancer-associated fibroblasts [25] since FAP is expressed during remodeling of the extracellular matrix, which occurs during wound healing [26] and in benign diseases like scleroderma and idiopathic pulmonary fibrosis [6]. While patients with comorbidities such as systemic sclerosis may not be the best candidates for treatment with systemic therapeutic FAPIs, biology-guided radiotherapy is potentially a good option for these patients, as the physician can identify specific FAPI-positive regions to treat while excluding FAPI-positive regions related to non-oncologic processes, provided they are sufficiently far away from the treatment targets.

Given that the NTS_FAPI_ was greater than the NTS_FDG_ in pancreatic cancer, liver cancer, and lung cancer, and the t-ratio for change in NTS between [^68^Ga]Ga-FAPI-04 and [^18^F]FDG was greatest for these disease sites, these disease sites deserve further investigation for BgRT. However, some liver tumors had a ring-like pattern of [^68^Ga]Ga-FAPI-04 that could make it difficult to develop an acceptable BgRT plan quality for certain patients. For pancreatic cancer, the NNA_FAPI_ was also much higher than the NNA_FDG_ since pancreatic cancer contains a high amount of desmoplastic stroma composed of cancer-associated fibroblasts. Since the NNA_FAPI_ is significantly higher than the NNA_FDG_ for pancreatic cancer, we may be able to reduce the injected activity of [^68^Ga]Ga-FAPI-04 and use fewer millicuries of [^68^Ga]Ga-FAPI-04 for each treatment. The difference in the GTV was also greatest in pancreatic cancer, suggesting that [^68^Ga]Ga-FAPI-04 may identify stromal elements of the tumor that do not light up on [^18^F]FDG PET-CT. Taken together, the significant difference between NTS_FAPI_ vs. NTS_FDG_ and the difference in GTV suggest that pancreatic cancer may have the greatest potential to be successfully treated with [^68^Ga]Ga-FAPI-04 guided BgRT (Figure 4).

We acknowledge that NNA and NTS alone may not fully capture the complexity of dose delivery in BgRT, especially in the context of patient motion. While our study aimed to provide a preliminary comparison of tracer suitability based on imaging-derived metrics, dosimetric and phantom validation would strengthen the conclusions. Future work involving motion phantoms and dose simulations to validate BgRT feasibility is being planned.

While this study focused on [^68^Ga]Ga-FAPI-04, it is important to acknowledge that other FAP-targeting tracers, such as [^68^Ga]Ga-FAPI-46 [27,28,29] and [18F]F-FAPI-74 [30,31], may exhibit different pharmacokinetics, tumor retention profiles, and background uptake characteristics. These differences could influence their suitability for BgRT applications, particularly in terms of signal persistence, biodistribution, and image contrast. For example, [^18^F]F-FAPI-74 benefits from a longer half-life and higher production yield, which may enhance workflow scalability. Future studies comparing these tracers head-to-head in the context of BgRT are warranted to determine the optimal agent for specific tumor types and clinical scenarios.

## 5. Conclusions

In conclusion, our work suggests that [^68^Ga]Ga-FAPI-04 represents a novel radiotracer for real-time BgRT in multiple cancer types, specifically in gastrointestinal and thoracic malignancies. This innovative approach could incorporate new imaging markers from nuclear medicine into radiation treatment planning and delivery to open a whole new paradigm in radiation oncology. Additional studies are needed to determine which patient populations will benefit the most from this approach.

## Figures and Tables

**Figure 1 cancers-17-03648-f001:**
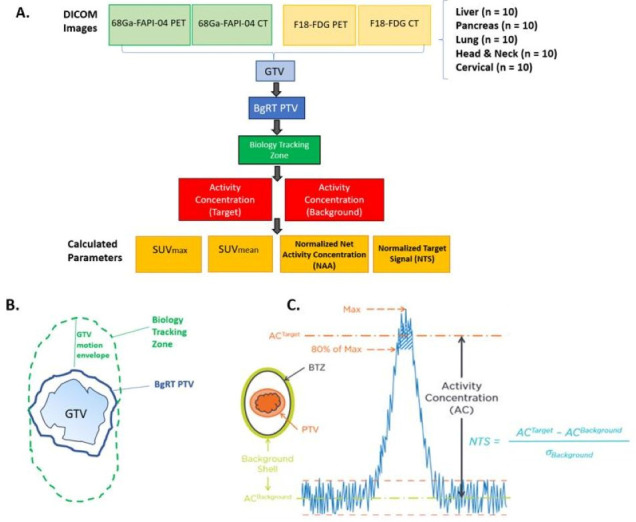
Schematic of methods and definitions for [^18^F]FDG vs. [^68^Ga]Ga-FAPI-04 PET study. (**A**). Study design: PET-CTs were obtained for 50 patients with liver, pancreas, lung, head and neck and cervical cancers using both [^18^F]FDG and [^68^Ga]Ga-FAPI-04 (*n* = 10 patients per disease site with 4 DICOM image sets each (PET and CT for both tracers). The gross tumor volume (GTV) was contoured on the [^18^F]FDG and [^68^Ga]Ga-FAPI-04 PET-CT images. The following parameters were then measured or calculated: GTV (in cc), SUVmax, volume of GTV receiving 80% of SUVmax, SUVmean (mean SUV in the volume receiving 80% of SUVmax), activity concentration of target, activity concentration of background, normalized net activity concentration (NNA), and normalized target signal (NTS). (**B**). Generating the biology tracking zone (BTZ): To create the BgRT PTV, the GTV was contoured, and a 3–5 mm margin was added based on disease site (see Appendix A for more info). To create the biology tracking zone (BTZ), a “GTV motion envelope” was added, incorporating each disease site’s average motion in all directions. (**C**). Normalized target signal (NTS) was defined as [AC (Target)−AC (Background)]/(Standard Deviation of Background). AC (Activity concentration) of the target was defined as mean AC (in kBq/mL) in the volume of GTV receiving 80% of SUV max. AC (activity concentration) of the background was defined as the mean AC (in kBq/mL) in the background shell (defined as a 3 mm margin around the BTZ).

**Figure 2 cancers-17-03648-f002:**
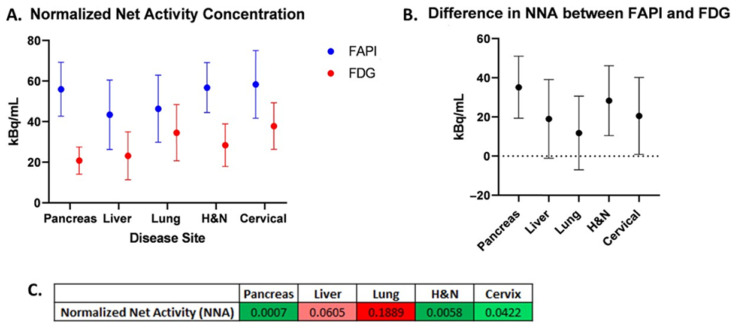
Normalized net activity concentration (NNA) for [^68^Ga]Ga-FAPI-04vs. [^18^F]FDG. (**A**) Comparison of mean NNA_FAPI_ vs. NNA_FDG_ for each disease site. (**B**) Difference in mean NNA_FAPI_ and NNA_FDG_ for each disease site. Error bars represent 95% confidence intervals. (**C**) Table of *p*-values from two-sided paired *t*-tests comparing NNA for [^68^Ga]Ga-FAPI-04 and [^18^F]FDG (green shades correspond to *p* < 0.05; red shades correspond to *p* > 0.05).

**Figure 3 cancers-17-03648-f003:**
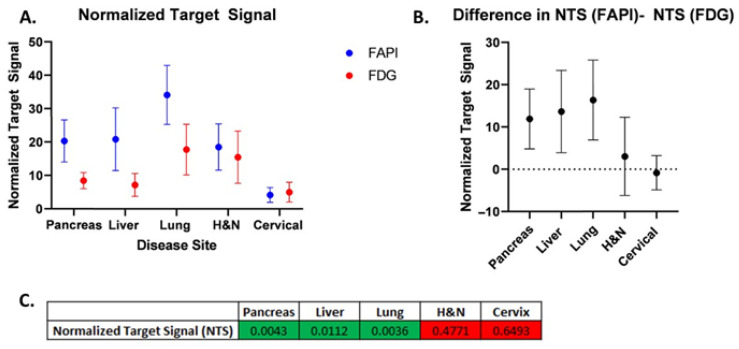
Normalized target signal (NTS) for [^68^Ga]Ga-FAPI-04vs. [^18^F]FDG. (**A**) Comparison of mean NTS_FAPI_ vs. NTS_FDG_ for each disease site. (**B**) Difference in mean NTS_FAPI_ and NTS_FDG_ for each disease site. Error bars represent 95% confidence intervals. (**C**) Table of *p*-values from two-sided paired *t*-tests comparing NTS for [^68^Ga]Ga-FAPI-04 and [^18^F]FDG (green shades correspond to *p* < 0.05; red shades correspond to *p* > 0.05).

**Figure 4 cancers-17-03648-f004:**
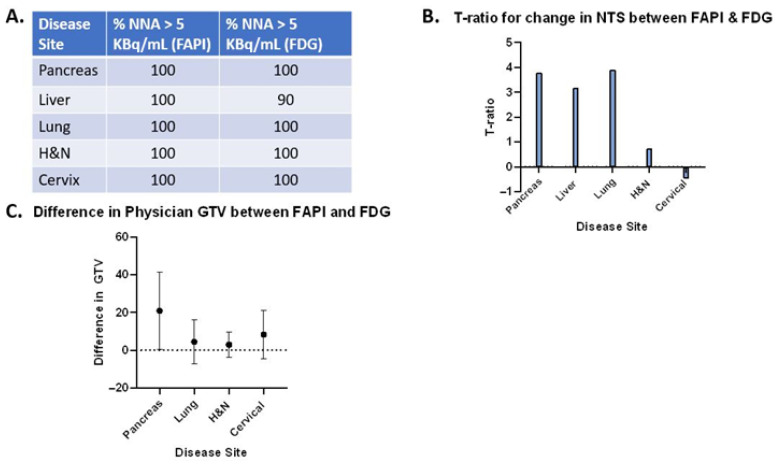
“Synthesis Matrix”, providing an overall comparison of [^68^Ga]Ga-FAPI-04 vs. [^18^F]FDG PET-CT. (**A**). Table summarizing the percentage of patient cases with NNA*_FAPI_* greater than 5 kBq/mL and the percentage of patient cases with NNA*_FDG_* greater than 5 kBq/mL. (**B**). t-ratio for standardized mean change in NTS between [^68^Ga]Ga-FAPI-04 and [^18^F]FDG for all disease sites. (**C**). Difference in gross tumor volume (GTV) in cc between [^68^Ga]Ga-FAPI-04 and [^18^F]FDG. Note: Liver contours were removed from this analysis because the volumes of some liver lesions were much greater due to the [^68^Ga]Ga-FAPI-04-avid rim around a central necrotic core. Error bars represent the standard deviation. Appendix A contains additional data tables, patient-level metrics including injected dose, uptake time, and tumor volumes for both tracers.

**Figure 5 cancers-17-03648-f005:**
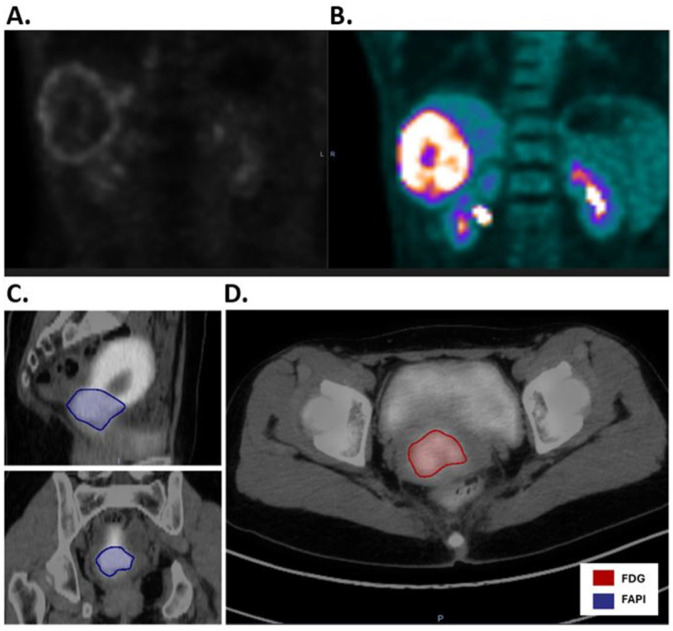
Comparison of [^68^Ga]Ga-FAPI-04PET-CT and [^18^F]FDG PET-CT for liver and cervical cancers. (**A**). [^68^Ga]Ga-FAPI-04 PET-CT of liver cancer reveals rim-like expression of [^68^Ga]Ga-FAPI-04 uptake surrounding the necrotic core of the tumor. (**B**). As a comparison, [^18^F]FDG PET-CT reveals an [^18^F] FDG-avid tumor in the liver, as well as [^18^F]FDG uptake in the bilateral kidneys. (**C**). Sagittal and coronal images of [^68^Ga]Ga-FAPI-04 PET-CT of cervical cancer illustrate a [^68^Ga]Ga-FAPI-04 positive GTV (blue) as well as significant [^68^Ga]Ga-FAPI-04 signal in the normal uterus. (**D**). As a comparison, the axial image of [^18^F]FDG PET-CT shows the GTV (red) with less background signal in the uterus.

## Data Availability

Research data are stored in an institutional repository and will be shared upon request to the corresponding author.

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
