# Peer review of "Comparative Evaluation of [68Ga]Ga-Fibroblast Activation Protein Inhibitor vs. [18F]FDG as a Novel Radiotracer for Biology-Guided Image Radiotherapy"

_cancers, 2025, doi:10.3390/cancers17223648_

Round 1
Reviewer 1 Report
Comments and Suggestions for Authors
This research article compares the [68Ga]Ga-FAPI-04 vs [18F]F-FDG tracers for BgRT using NNA and NTS as an imaging parameters. While FAPI often known to have higher tumor uptake. Studying these two parameters alone does not guarantee the effective dose delivery in moving patient, no dosimetric or phantom studies are provided. Moreover, the SI is missing, experimental details like injected dose, uptake time, tumor lesions volumes, PET parameters and standard nomenclature of radiopharmaceuticals is missing too (PMID:29074076). Given that the manuscript is not suitable in its current form and authors may revise and try new submission.
Reviewer 2 Report
Comments and Suggestions for Authors
Comment
This paper by Qiu et al. explored the difference of 68Ga-FAPI-04 compared with 18F-FDG for biology-guided radiotherapy (BgRT) across several tumor types. 68Ga-FAPI-04 showed better performance compared to 18F-FDG for BgRT, which had potential impact for pancreatic cancer. I would suggest the publication of the manuscript in cancers after addressing the following comments:
- Please replace all figures (Figure 1-5) with high resolution.
- Please show the chemical structure of 68Ga-FAPI-04 and 18F-FDG to show the difference between them.
- Please provide an additional explanation of why NNA for 68Ga-FAPI-04 and 18F-FDG in liver and lung shows no obvious difference.
- Please provide figures to quantify the intensity of the indicated regions shown in Figure 4A-D to compare the differences between PET tracers in different cancers (Please also provide the significant difference).
Reviewer 3 Report
Comments and Suggestions for Authors
General comments
The authors compared the use of F-18-FDG and Ga-68-FAPI-04 for radiation planning in 50 patients with 5 different tumors. The topic is interesting. However, the methodology used for the data evaluation is not state of the art.
Specific comments
- Please use the correct nomenclature for radiopharmaceuticals, e.g. [68 Ga]Ga-FAPI-04 and [18F]FDG.
- Please replace “PET radiotracer emission” by “PET radiotracer uptake”.
- Line 30: “FDG…it is taken up by all proliferating cells”. This is not completely true. Please change into “FDG……it is taken up by all viable tumor cells as well as inflammatory cells”. FDG is not a proliferation marker, like FLT.
- Line 40-41: “Normalized Net Activity Concentration (NNA) and Normalized Target Signal (NTS)”. Please provide more informations about the theoretical background of these calculations. Who introduced these values in radiation therapy planning? Which software has been used for these calculations? Is this software provided by the manufacturer?
- Line 53: “..and facilitates…” should be replaced by “..and may facilitate…”
- Line 54: “…PET avidity….” should be replaced by “…PET radiotracer avidity…”
- Line 58: …” producing significant background in the brain…” should be replaced by “…resulting in significant …”
- Line 61: “…in generating new radiotracers…” should be replaced by “..in developing new radiotracers…”
- Line 71: “In addition, FAPI PET-CT is better than 18F-FDG PET-CT for detecting metastatic lymph nodes and distant metastases.” Please add “as stated by the studies of….”.
- Line 82-84 and Ln 147-150: “the Normalized Net Activity Concentration (NNA) (goal > 5 kBq/mLto deliver BgRT), and the Normalized Target Signal (NTS) (goal > 2.7 for BgRT treatment planning & > 2.0 for BgRT delivery) [8].” This reference is only an abstract based on 5 patients with lung tumors and 4 patients with bone tumors. Please provide a more valid reference for the NNA and NTS values.
- Line 124: please explain ITV.
- Line 116-133: The cutoff of SUVmax threshold used for contouring is confusing. When did the authors use a 40% SUVmax isocontour (ln 116-119) and when 80% (ln 132)?
- Ln 133-144? It is not clear to me why the authors did not use SUV instead of activity concentrations. SUV are normalized to the body weight and injected dose and also include decay correction. The calculations described here are questionable to my opinion and not state of the art.
- 2 and 3: SUVmax and SUVmean as well as tumor-to-background ratios (TBR) should be presented here.
- SUVmax and SUVmean as well as tumor-to-background ratios (TBR) should be presented for every of the studied tumors. NNA and NTS values are non-standard.
- A comparison between FAPI-04 and other FAPIs like Ga-68-FAPI-46 and F-18-FAPI-74 is missing.
Round 2
Reviewer 1 Report
Comments and Suggestions for Authors
By including key parameters and acknowledging study limitations the MS has been greatly improved. I appreciate the authors by addressing the comments thoughtfully. Its ready to accept after minor editorial check.
Reviewer 3 Report
Comments and Suggestions for Authors
No further comments.
Comments on the Quality of English LanguageThere are still some spelling errors, like introduction second line "methiodmay"....or discussion last lines "[18F]F-FAPI-74" instead of [18F]FFAPI-74 etc.